# Evaluation of an Arabic Chatbot Based on Extractive Question-Answering Transfer Learning and Language Transformers

**Tahani N. Alruqi and Salha M. Alzahrani \*** 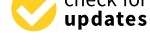

Department of Computer Science, College of Computers and Information Technology, Taif University, P.O. Box 11099, Taif 21944, Saudi Arabia; s44280557@students.tu.edu.sa
\* Correspondence: s.zahrani@tu.edu.sa

**Abstract:** Chatbots are programs with the ability to understand and respond to natural language in a way that is both informative and engaging. This study explored the current trends of using transformers and transfer learning techniques on Arabic chatbots. The proposed methods used various transformers and semantic embedding models from AraBERT, CAMeLBERT, AraElectra-SQuAD, and AraElectra (Generator/Discriminator). Two datasets were used for the evaluation: one with 398 questions, and the other with 1395 questions and 365,568 documents sourced from Arabic Wikipedia. Extensive experimental works were conducted, evaluating both manually crafted questions and the entire set of questions by using confidence and similarity metrics. Our experimental results demonstrate that combining the power of transformer architecture with extractive chatbots can provide more accurate and contextually relevant answers to questions in Arabic. Specifically, our experimental results showed that the AraElectra-SQuAD model consistently outperformed other models. It achieved an average confidence score of 0.6422 and an average similarity score of 0.9773 on the first dataset, and an average confidence score of 0.6658 and similarity score of 0.9660 on the second dataset. The study concludes that the AraElectra-SQuAD showed remarkable performance, high confidence, and robustness, which highlights its potential for practical applications in natural language processing tasks for Arabic chatbots. The study suggests that the language transformers can be further enhanced and used for various tasks, such as specialized chatbots, virtual assistants, and information retrieval systems for Arabic-speaking users.

**Keywords:** Arabic; chatbot; transfer learning; AraBERT; CAMeLBERT; AarElectra (Generator/Discriminator); AraElectra-SQuAD

## 1. Introduction

Computer programs called "chatbots" mimic human speech. They run based on artificial intelligence (AI) technologies and natural language comprehension. Customer service, e-commerce, healthcare, banking, gaming, education, travel, tourism, and other sectors utilize chatbots extensively. English chatbots have been the subject of extensive investigations since the 1960s. In 1966, the first chatbot, named Eliza, was created [1]. By returning the user's phrases in the interrogative form, Eliza acted as a psychotherapist, which was an inspiration for the creation of later chatbots. Despite its limited communication capabilities, the Eliza chatbot employed pattern matching and response selection methods based on template sets; thus, it could discuss a restricted range of subjects. Additionally, this early chatbot was unable to maintain lengthy dialogs or gather context from the discourse. These days, a broad range of chatbots have been developed, which can be divided into two categories: those that employ techniques used in traditional rules-based chatbots and intelligent chatbots. In traditional rule-based chatbots, a set of rules and templates of set answers are used with pattern matching and response selection methods. On the other hand, intelligent or smart chatbots are developed based on AI, natural language

processing (NLP), and natural language understanding (NLU) methods and can respond in a human-like manner. To respond to queries and access a sizable body of knowledge, machine learning (ML) and deep learning have been widely used in chatbot research.

Arabic chatbots are uncommon because of the nature and intricacy of the Arabic language. The use of chatbots in Arabic is a relatively recent trend, but it shows a great deal of promise. Arabic chatbots have significant potential and, given the remarkable rise in digital technology, many companies and organizations are keen to use them, aiming to revolutionize how users and consumers who speak Arabic interact with their services. Modern NLP and NLU, which enable the interpretation and creation of computerized methods for natural languages, have enabled the development of Arabic chatbots. As a result, it has become easier for different sectors, businesses, and companies to create chatbots that can comprehend and answer questions in Arabic. Examples of research works related to Arabic chatbots include the design of frameworks for building and utilizing them in different applications [2–4]. To build an Arabic chatbot, one needs to comprehend the language, culture, and intricacies of Arabic. In addition to the technical aspects of building a chatbot, the cultural and social contexts of the regions targeted by the chatbot should be understood to enable the chatbot to engage in meaningful discussions with people. An intelligent chatbot should also be familiar with the many dialects and nuances of the Arabic language and must be able to comprehend the cultural context of interactions. Consequently, rather than relying solely on pre-written responses, the chatbot should be able to converse with customers naturally. For instance, there are several dialects and accents in Arabic; therefore, the chatbot must be able to comprehend them while interacting with people. One important requirement is that the chatbot can understand other languages and translate them into Arabic.

Researchers have devoted significant amounts of time and effort to developing frameworks and structures for chatbots in various applications. The incorporation of advanced technologies, such as natural language processing and question–answer databases, enhances the conversational abilities of chatbots. Different types of question-answering (QA) systems are used in building chatbots; these exhibit variations in the ways they produce answers. One such type is extractive QA, where the system retrieves the answer from a given context and presents it to the user using BERT-like models. To elaborate, extractive QA chatbots involve searching through a large corpus of information and identifying one correct answer. This method is important for close-domain specialized chatbots that aim to provide specific answers. Also, extractive QA models use transfer learning, which provides a pre-trained model with a lot of data and uses self-attention to uncover relationships between words in a sequence that are dependent on one another [5,6]. Another type is generative QA models, which generate a textual answer from scratch using AI-based generation models. Examples such as Bing Chat (Microsoft) and ChatGPT 3.5 and 4.0 (OpenAI) aim to answer open-ended questions by generating long sequences of answers [7,8]. The rise of generative AI models, their current limitations, and their future directions have been investigated in several studies [9–11].

Few research studies have investigated the use of deep neural learning and NLP methods in Arabic chatbots [12–15]. To the best of our knowledge, none of these studies have investigated both extractive QA and transfer learning methods using language transformers to evaluate Arabic chatbots. Therefore, the contributions of this study are threefold:

- First, BERT-like language transformers that were pre-trained on large collections of Arabic were explored for use in an Arabic QA chatbot.
- Second, the transfer learning method was investigated for Arabic chatbots using Arabic language transformers, namely, AraBERT, CAMeLBERT, AraElectra-SQuAD, and AarElectra (Generator/Discriminator).
- Third, the proposed methods were evaluated with NLP evaluation metrics and by using two Arabic QA datasets, which demonstrated that Arabic chatbots can meaningfully understand the conversations' contexts and respond naturally.

The rest of this study is organized as follows. In Section 2, a comprehensive literature review about chatbots is provided, including chatbot system architectures, applications, and the methods used in chatbot research, namely rules-based, corpus-based, and retrieval-based methods, and extractive versus generative AI methods. In Section 3, the proposed methodology of this work is explained, including the problem formulation and general framework, the transformers used for the Arabic language, and the methods of transfer learning proposed in this study. The dataset exploration, resources, tools, and evaluation metrics are given in Section 4. The experimental results and discussion are presented in Section 5. Finally, the concluding remarks and directions for future Arabic chatbot research are given in Section 6.

## 2. Literature Review

In this section, we review chatbot studies from a data science perspective, covering the years between 2019 and 2023. Background knowledge or theoretical concepts were taken from references from before this range of years.

### 2.1. Chatbot System Architecture and Applications

The general architecture of chatbot input and response generation proposed by [16] consists of the user message analysis component (UMAC), which analyzes the query or request sent by the user to determine the entity and intent categorization, and the dialog management component (DMC), which handles all textual and voice messages that are sent back and forth (questions, requests, and responses). A chatbot must choose its next course of action once it has reached the best interpretation it can. It has several options, including asking for clarification, requesting more background information, remembering what it has learned, and waiting to see what occurs next. Once the request is comprehended, the chatbot seeks data from a database or through API calls from the backend. The knowledge base utilized by rule-based chatbots comprises handwritten replies to user inputs, whereas the generative model uses natural language generation to provide replies. At the user end, the chatbot outputs the message to answer the request.

A variety of factors are used to classify chatbots. These criteria may include the way in which the chatbots are implemented, the knowledge domain they are used in, how they are used, and the methodologies used to generate responses. Depending on how a chatbot interacts with its input and output, it can be classified as a text- or speech-based conversational dialog. While open-domain chatbots may converse about a variety of topics, closed-domain chatbots concentrate on specific knowledge areas and may struggle to address unrelated issues. Task-oriented chatbots are created to help users complete specific tasks in a particular domain, whereas non-task-oriented chatbots encourage user participation across several topics and act as informational chatbots. Chatbots are categorized depending on their response generation approaches into rules-based, corpus-based, retrieval-based, and extractive/generative AI-based chatbots. Rules-based chatbots are designed to respond to specific queries and can effectively produce answers based on established guidelines. However, they are unable to handle inputs with spelling and grammatical errors and can only address the most recent message entered. Corpus-based and retrieval-based methods can handle a wider range of questions than rules-based methods as they employ statistical language modeling. Recently, extractive/generative AI-based chatbots have been powered by AI that is designed to communicate with users, much as actual people do, by using NLP, NLU, ML, deep learning, and other techniques to comprehend the user's intent, with the ability to remember context and diction. The process of training AI-based chatbots begins with datasets that are collected for the purposes of the chatbot, with preprocessing required to clean the data; this is followed by designing and training the model, which can be either a machine learning or a deep learning model, and finally testing and evaluating the model [17].

Chatbots are used in a wide range of industries and applications, such as customer service, e-commerce, healthcare and medical applications, education, and language learn-

ing [15], as summarized in Table 1. In customer service, the chatbot is used to respond to frequently asked questions, to resolve problems, and to deliver information; in this way, chatbots are utilized to provide customer care [18–20]. For online shopping [21,22] and in the circular economy [23], e-commerce chatbots offer product recommendations, respond to inquiries about specific items, and guide users around a website. Furthermore, healthcare, pharmaceutical, and medical diagnosis chatbots can help with appointment scheduling, giving medical advice and information, and even classifying symptoms [24–29]. Educational chatbots are used to support students, help with registration and enrollment, and provide information about courses and activities [30–34]. Socially, chatbots can be used in education for parental practice [35]. Finally, for aiding in language learning, chatbots can be used to help users learn a second language, correct grammar, and aid in writing/speaking improvements [5,36].

**Table 1.** Summary of research works on chatbots applications including customer service, e-commerce, healthcare and medical diagnosis, education, and language learning.

| Industry/Application | Description | Pros. | Cons. | Ref. |
|---|---|---|---|---|
| **Customer service** | Sentiment and intent analysis and emotion recognition in customer service chatbots | supports emotions and sentiment analysis | accuracy is not high | [18] |
| | Goal-oriented conversation management bootstrapping | transfers learning and improves accuracy | low data and domain specific | [19] |
| | Chatbots and voice assistants: digital transformers of the company and customer service | has an extensive literature review | - | [20] |
| **e-Commerce, Economy, and Telecom** | Engagement with chatbots versus augmented reality interactive technology in e-commerce | tests consumers' attitude and engagement | cannot be generalised | [22] |
| | Information technology telecom chatbot | can be integrated into other online platforms | rule based chatbot | [37] |
| | Use and design of chatbots for the circular economy | analysis of five existing chatbots | no practical implementation | [23] |
| **Healthcare and Medicaldiagnosis** | Ask Rosa: digital genetic conversation chatbot about hereditary breast and ovarian cancer | extensive user and formal usability testing | manual building of the database | [24] |
| | AI-Powered health chatbots general architecture using NLP and NLU | gives response from pre-formatted data | no practical implementation | [25] |
| | Chatbot for disease prediction and treatment recommendation | great for daily check-ups | no practical implementation | [26] |
| | Mental healthcare chatbots | assists mental healthcare using deep learning | low datasets for this domain | [27,28] |
| | Design of an educational chatbot in radiotherapy | disseminates topics in radiotherapy | limited data | [29] |
| **Education** | Highly adaptive educational chatbot | can detect the student's intent | no practical implementation | [38] |
| | NEU-chatbot: chatbot for admission of National Economics University | students get daily updates instantly | - | [21] |
| | Educational and smart chatbots for colleges and universities | uses NLP and ready-to-use platforms | no detailed explanation of the methods | [30–34] |
| | Designing a chatbot for helping parenting practice | solves problems encountered by novice parents | no detailed explanation of the methods | [35] |
| **Language learning** | Chatbot assistant for English as a second language learners | used in real-world applications | no evaluation methods used | [5,36] |

## 2.2. Chatbot Methods

### 2.2.1. Rules-Based Methods

Traditional and early research work on chatbots employed rules-based methods [39–41], which utilized a collection of predetermined human-created rules, applied in a hierarchy,

to transform the user input into an output. Instead of creating a brand-new answer, the rules divide the input into a series of tokens to look for patterns and produce a response. Although this strategy is simple and straightforward to develop, it confines responses to inputs that fall within the stated rules only and fails to address questions/queries that are not in the collection [42].

### 2.2.2. Corpus-Based and Retrieval-Based Methods

Developing after traditional rules-based chatbots, corpus-based and retrieval-based methods have been widely employed; they work by utilizing a corpus or knowledge base using a statistical language approach [42]. The majority of chatbots in this category employ information retrieval to extract a candidate response from the corpus based on heuristics approaches. In these methods, both the input and the context are considered by recognizing keywords to provide the best answer from the corpus/knowledge base, as opposed to employing predetermined criteria.

### 2.2.3. Extractive-Based and Generative-Based Methods

AI-based solutions do not require any prepared replies; instead, the AI creates the responses depending on the context of the dialog. By considering both recent and past user interactions, the chatbot tries to come up with a fresh response. It is necessary to gather a sizable training set, which can be challenging. Due to the real-time nature of this method, response failures are quite likely [42]. AI-based chatbots can be classified into extractive and generative QA systems based on the method used to provide the responses. Extractive QA chatbots are designed to answer questions by extracting the relevant information from a given passage of text. They employ algorithms to identify key pieces of text in the passage that are most likely to contain the answer and then extract those pieces of text to provide an answer to the question. Generative QA chatbots, on the other hand, generate responses to questions from scratch using AI generation models that are trained on large collections. While extractive QA models tend to produce more accurate answers, generative models can generate entirely new responses that may provide a deeper understanding of the given text. Both types of QA models have their own strengths and weaknesses and are used in different applications depending on the desired outcome. The following subsections discuss various methods used in generative and extractive chatbot research, including NLP, NLU, ML, recurrent neural networks (RNNs), long short-term memory networks (LSTMs), gated recurrent units (GRUs), encoder–decoder, sequence-to-sequence (Seq2Seq), reinforcement, and transfer learning.

**NLP** methods rely on computational linguistics and statistical modeling with textual data. NLU, as a part of NLP, analyzes texts and voices using syntactic and semantic thesauruses and knowledge bases. While semantics relates to the sentence's intended meaning, syntax refers to a sentence's grammatical structure. A general architecture for chatbots that combines dialog and communication components with NLU, as well as expert components, was developed based on deep learning [13,25]. Their AI-powered chatbots enabled interactions with users in a more human-like manner while providing accurate answers to their questions related to either open domains [13] or closed domains, such as healthcare [25] and education [21,30].

**ML methods** have been used in several research works for chatbots [12,17,26,37,43–47]. ML-chatbot-related research works were investigated and reviewed up until 2020 by Suta et al. [15]. ML was utilized to understand the relationships and intentions in queries using simple logistic regression and an iterative classifier optimizer; it achieved 97.95% accuracy in predicting users' intentions, a value higher than that achieved by other classifiers [43]. A support vector machine (SVM) algorithm for predicting the health status of users was integrated with Google API for speech-to-text and text-to-speech conversions [44]. The k-nearest neighbor method (KNN) was used to extract symptoms from conversations to provide diagnoses and therapy recommendations [26]. An SVM-trained model was examined in relation to women's potential to develop depression, anxiety, and hypomania.

To give users spiritual support and medical guidance, the users' mental health data were gathered and assessed in real time using a chatbot paired with the psychological test scale for additional diagnoses [27]. A chatbot framework using ML was proposed that aimed to diagnose and solve technical issues using extracted data from technical support tickets in HP and Microsoft in Arabic [12].

**RNNs** are a subset of artificial neural networks that can handle sequential inputs by passing information from one step in a sequence to the next via loops [6]. Several research works have used RNN methods for different purposes in chatbots, such as classifying the users' intentions [48], detecting the users' emotions and feelings [18], and handling long conversational dialogs [49]. RNNs were able to comprehend similar-sounding phrase variants and enhance a conversation intent classification model, which obtained 81% accuracy [48]. Another study collected data based on discussion and sentiment analysis and used recent talks as the input to RNNs to classify the feelings; it obtained 0.76 precision accuracy [18]. Bidirectional RNN and an attention model were used to generate responses to long queries (more than 20–40 words) [49].

**LSTMs**, as a type of RNN, were created to capture long-term dependencies by retaining and forgetting information over a range of time steps [6]. Different studies have utilized LSTMs in conversational dialogs to provide a better understanding of the context of a conversation and more accurate and coherent responses from the chatbot [31,50–53]. A study showed that the performance of a chatbot was improved by using an ensemble of LSTM networks, rather than a single LSTM model, trained to learn long-term dependencies and the relationships between events that occur over a prolonged period [50]. The impact of context learning on the chatbot's overall performance using LSTM with a metaphorical approach was examined [51]. Interactive chatbots based on the LSTM and NLP algorithms were developed for teaching, serving students, and responding to queries posed by pupils [31,52]. Furthermore, a study by Anki, Bustamam, Al-Ash, and Sarwinda [53] proposed a bidirectional LSTM, known as BiLSTM, to analyze input sequences both forward and backward, enabling it to recognize long-term connections between a sequence's component parts. The model's performance was assessed using many datasets, and the chatbot achieved an average accuracy of 0.99.

**GRUs** are a kind of RNN that employ gating mechanisms to regulate the information flow to and from memory cells. With fewer parameters and processes, a GRU unit often performs as well as or better than an LSTM [6]. One study utilized GRUs as chatbots in web interfaces, which proved that the performance of the GRU was better at answering questions than BiLSTM using the Facebook bAbi dataset [51].

**Encoder–Decoder** structures contain an encoder, which transforms input data into the internal representation of the network using a fully connected hidden bottleneck layer, whereby its activation vector is considered the internal state. On the other hand, the decoder attempts to rebuild the input from the internal data model of the network [6]. Several research works have used encoder–decoder methods and attention mechanisms for chatbots and conversational dialogs [17,38,54]. These studies are intended to improve the response generation for user queries as well as the experiences and interactions of humans with chatbot technology.

**Seq2Seq** models are based on encoder–decoder architectures and are widely used in NLP tasks such as machine translation, text summarization, QA, text generation, and more, whereby the input is a sequence of data and the output is another sequence. Seq2Seq models consist of an encoder network that converts the input sequence into a fixed-length vector representation and a decoder network that decodes the representation into the output sequence [6]. A study proposed midoBot, a Seq2Seq-based deep learning Arabic chatbot that textually converses with others on common conversational subjects [55]. Another study developed a chatbot using a Seq2Seq encoder–decoder architecture trained on brain disorders and mental illness data that successfully reacted sympathetically to people with mental illnesses [28].

**Reinforcement learning** models train the machine by using incentives and penalties. The model's objective is to maximize its overall rewards during training, and it accomplishes this by acting in ways that can alter the environment [6]. Reinforcement learning has been utilized in chatbots [56], wherein the reward method enables the chatbot to distinguish between correct and incorrect responses. With the use of deep reinforcement learning algorithms, this chatbot can recognize the tone of a question and respond appropriately. Q-learning, a deep Q-neural network (DQN), and distributional reinforcement learning with quantile regression methods were utilized in the suggested system (QR-DQN), and the performance of each method was investigated and evaluated. Ensemble-based deep reinforcement learning for chatbots with sentence clustering and dialog clustering was developed and trained on raw dialog textual data only, without any manually labelled data [57]; the researchers concluded that ensemble chatbot agents were highly correlated with human-rated data. A recent study [58] combined different reward functions using an encoder–decoder model and reinforcement learning. The study showed competitive results on the SQuAD dataset for the extractive QA chatbot.

**Transfer learning** leverages the knowledge learned by a pre-trained network on a large dataset to a new related problem [6]. In order to counteract the negative consequences of the limited availability of data for a chatbot in a specific domain, several research works have sought to modify a model that was originally developed for one task and apply it to a related task, specifically, for chatbots [5,19,59,60]. This was achieved by enhancing a pre-existing language model and fine-tuning it on a specific set of conversational data, such as medical consultations or customer service encounters. In this regard, chatbots employ the broad linguistic comprehension skills learned from the pre-trained model while simultaneously learning the specific terms and language used in a closed domain. The creation of chatbots using transfer learning-based strategies involves the fine-tuning of pre-trained transformer models such as BERT or GPT-2 and pre-trained embeddings including GloVe and ELMo, together with the neural network architecture using available conversational data [5]. A method utilizing transfer learning improved the chatbot's performance by 20% for open domains and more than doubled the improvement for closed domains [19], wherein the most favorable outcomes arose when the transfer learning technique was merged with complementary processing techniques such as warm-starting. Two recent studies investigated the use of transfer learning to enhance the ranking of responses in extractive-based QA chatbots [60], and to improve the response generation [61].

### 2.3. Discussions and Research Gap

The trend of using both extractive-based and generative-based AI methods for chatbots is gaining popularity. Many methods of NLP, as well as NLU [13,21,25,30], ML [24,27,30,36,37,62], and deep learning, including RNN, LSTM, GRU, encoder–decoders, Seq2Seq [5,17,21,28,31,38,48–55,57], and reinforcement learning [56,57,60,61], have been successfully applied in research related to chatbots and conversational dialogs. Table 2 summarizes several research works that were conducted using these models, the datasets used, and their advantages and disadvantages from our perspective. The table also shows a comparison between the levels of accuracy obtained by the different models examined in these studies. One of the latest directions in chatbot research is the use of transfer learning by some research works [5,19,59,60], whereby the knowledge leveraged from a pre-trained model helps the chatbot to understand and comprehend the context while interacting with people. As other researchers have investigated the use of ML, NLP, and deep learning methods in Arabic chatbots [12–14,32,33,63], we believe that modern NLP, which enables the interpretation and creation of computerized methods for natural languages, along with the recent trends in transfer learning, has the potential to aid the development of Arabic chatbots. Our literature review confirms that the use of transfer learning with recent language transformers will open up new directions for more specialized extractive and generative QA in Arabic chatbots. Therefore, this research aims to bridge this gap, moving towards developing Arabic chatbots that understand the conversational context and behave naturally, as humans

do, by conducting a thorough implementation of BERT-like Arabic transformers, as well as a comparison and evaluation, using transfer learning on existing QA datasets.

**Table 2.** Comparison of research works in chatbots and conversational dialogs, including NLP, NLU, machine learning, and deep learning, including RNN, LSTM, GRU, Encoder-Decoder, and Seq2Seq, with the datasets, accuracy results, and pros and cons used in each study.

| Method | Description | Dataset | Acc. | Pros. | Cons. | Ref. |
|---|---|---|---|---|---|---|
| NLP NLU | AI-powered healthcare chatbots NLU chatbot framework | - | - | utilizes NLP, NLU, NLG, deep learning | inaccurate data decrease accuracy | [13,21,25,30] |
| ML | Acceptance of chatbot based on emotional intelligence through machine learning algorithm | international students with experience in using chatbot | 97% | TAM and EI theory to predict users' intentions | data limited to international students, difficult to interpret | [43] |
| | An improved chatbot for medical assistance using machine learning | various sources: medical journals, online forums, and websites | 93% | streamlines medical processes and save time | SVM's accuracy may not be perfect | [44] |
| | Chatbot for disease prediction and treatment recommendation using machine learning | comprised of patient data, medical history, and symptoms | - | alternative to hospital visits-based diagnosis | not as accurate as traditional hospital visits | [26] |
| | Supervised machine learning chatbots for perinatal mental healthcare | pregnant women, newborns, and their families | - | reduces barriers and helps clinicians make accurate diagnoses | cannot accurately detect subtle changes in mental health | [27] |
| | A novel framework for Arabic dialect chatbot using machine learning | extracted IT problems/solutions from multiple domains | | accuracy, response time | no explanation of how ML was employed | [12] |
| RNN | Intents categorization for chatbot development using Recurrent Neural Network (RNN) Learning | university guest book available from its website | 81% | understands variations in sentence expression | requires big data, difficult or expensive to implement | [48] |
| | Conversations sentiment and intent categorization using context RNN for emotion recognition | conversations inside a movie | 79% | successful in recognizing emotions in text-based dialogs | only uses a single dataset for testing the algorithm | [18] |
| | Deep learning with bidirectional RNN and attention model | Reddit dataset | - | performs English-to-English translation | No accuracy measured | [49] |

<div align="center"><b>Table 2.</b> <i>Cont.</i></div>

| Method | Description | Dataset | Acc. | Pros. | Cons. | Ref. |
|---|---|---|---|---|---|---|
| **LSTM** | LSTM-based ensemble network to enhance the learning of long-term dependencies in chatbot | Cornell Movie Dialog Corpus | 71.59% | retains contextual meaning of conversations | - | [50] |
| | A metaphorical study of variants of recurrent neural network models for context learning chatbot | Facebook bAbi dataset | 96% | helps to create chatbots for web applications | only tests RNN models on a single dataset | [51] |
| | Natural language processing and deep learning chatbot using long-short term memory algorithm | conversations with users and assessments | - | understands questions and provide detailed answers | does not address accuracy and reliability | [52] |
| | AI based chatbots using deep neural networks in education | set of answer and question pairs | - | provides accurate and useful responses to student queries | incorrect/difficulty handling complex queries | [31] |
| | AI chatbot using deep Recurrent Neural Networks based on BiLSTM model | Cornell Movie Dialog Corpus | 99% | outperforms other chatbots in accuracy and response time | only compares with a few other systems | [53] |
| **GRU** | A metaphorical study of variants of Recurrent Neural Network models for a context learning chatbot | Facebook bAbi dataset | 72% | - | - | [51] |
| **Encoder-Decoder** | AI chatbot based on encoder-decoder architectures with attention | Cornell Movie Subtitle Corpus | - | improves the experience and interaction | lack of review of similar methods | [17] |
| | Behavioural chatbot using encoder-decoder architecture | - | - | increases replicability | focuses on mimicking fictional characters | [54] |
| | Highly adaptive educational chatbot using encoder-decoder framework for intent recognition | - | - | bidirectional transformer (CamemBERT) | no experimental evaluation | [38] |
| **Seq2seq** | Chatbot in Arabic language using Seq-2-Seq model. | ~81,659 pairs of conversations | - | uses common conversational topics | no detailed description of the dataset, making it difficult to replicate | [55] |
| | Mental healthcare chatbot using Seq2Seq Learning and BiLSTM | The Mental Health FAQ | - | assists mental healthcare | - | [28] |

**Table 2.** *Cont.*

| Method | Description | Dataset | Acc. | Pros. | Cons. | Ref. |
|---|---|---|---|---|---|---|
| **Transfer Learning** | Goal-oriented chatbot dialog management bootstrapping with transfer learning | - | - | overcomes low in-domain data availability | focuses on technical aspects not chatbot performance | [19] |
| | The design and implementation of English language transfer learning agent apps | English Language Robot | - | integrates recognition service from Google and GPT-2 | no comparison with existing chatbots for language learning | [5] |
| | Building chatbot using transfer learning: end-to-end implementation and evaluation | - | | shows fine-tuning and optimizing | no comparison evaluation | [59] |
| | Reranking of responses using transfer learning for a retrieval-based Chatbot | WOCHAT dataset Ubuntu dialogue dataset | | highest ratings from the human subjects | - | [60] |
| **Reinforcement Learning** | Evaluating the performance of various deep reinforcement learning | Cornell Movie-dialogs corpus and CoQA | - | comprehensive review of methods | difficult to compare to other approaches | [56] |
| | Ensemble-based deep reinforcement learning for chatbots | Chitchat data | - | training ensemble of agents improved chatbot performance | requires more training time | [57] |
| | Modeling extractive QA encoder-decoder reinforcement learning | SQuAD dataset | - | combines different reward functions | results need to be improved | [58] |
| | Exploring Bi-Directional Context for Improved Chatbot Response Generation | some generated samples | - | combines different models | qualitative evaluation | [61] |

## 3. Methodology

### 3.1. General Framework

Arabic chatbots, in general, can be created using a combination of ML, NLP, NLU, and/or transfer learning techniques, which can operate as shown in Figure 1. There are two sides to the chatbot operation: the user's question (i.e., the query), referred to as *Q*, and the bot's answer (i.e., the response), referred to as *R*. A given segment of textual data, referred to as context, *C*, is required to train the chatbot. The Arabic NLP is required to split the query into tokens (sentences, phrases, words, etc.), and the Arabic NLU is also required to utilize the meanings, nuances, and synonyms used in these tokens. ML is part of this framework, as we need to train the model with a sizeable dataset of questions and their contexts (*Q*, *C*, *R*). Deep learning can also be utilized to train the chatbot models with a set of questions and their contexts (*Q*, *C*) and it learns automatically to generate responses and interact naturally. To elaborate, the context "اسمي تهاني وآسكن في الطائف" is given to the model, with queries to be trained for by answering them. Then, if the user presents a

query or question to the chatbot, e.g., "أين تعيشين؟", where we use a different Arabic word to mean the same action of "living", the chatbot should be able to answer the question, "في الطائف". Instead of training the chatbot model from scratch, pre-trained language models, referred to as transformers, can be utilized with the transfer learning approach.

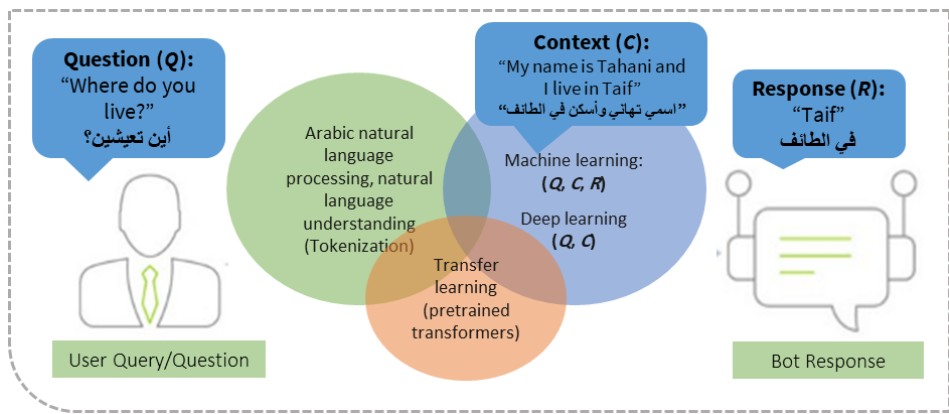

**Figure 1.** Operational framework of Arabic chatbot using transfer learning.

Question answering (QA) is the process of using a natural language processing model to automatically answer questions posed in natural language. Extractive QA involves interpreting the question, searching through a large corpus of information, and identifying the most relevant information to extract the correct answer. The goal of our method is to develop an Arabic chatbot based on extractive QA that can accurately and efficiently provide answers to questions posed in Arabic, similarly to how a human expert might provide answers. The methodology proposed for this research study is shown in Figure 2, which shows that the datasets for Arabic QA that are most closely related to this problem were utilized; they include hundreds of questions, as will be explained shortly. In order to train the models for extractive Arabic QA, we implemented transfer learning and fine-tuning on four sets of transformers, namely, the AraBERT, CAMeLBERT, AraElectra-SQuAD, and AraElectra (Generator/Discriminator) transformers with different variations and semantic embedding models, achieving state-of-the-art results for Arabic NLP problems. When implementing transfer learning, one dense layer and a softmax layer were added to fine-tune the AraBERT and CAMeLBERT pre-trained models because they were pre-trained as general text prediction models; meanwhile, only a softmax layer was added at the top of the AraElectra-SQuAD and AraElectra (Generator/Discriminator) transformers, as these were pre-trained for text discrimination and QA data. Finally, our fine-tuned models were then used to predict unanswerable questions and evaluate their performance using confidence and similarity metrics that are commonly used in NLP research.

*3.2. Details of Extractive QA and the Transfer Learning Method for Arabic Chatbots*

Extractive QA models involve creating a system that can automatically answer questions based on a given textual corpus. To build our extractive QA models, transfer learning was applied with the aim of evaluating and improving the performance of conversational dialogs in Arabic chatbots. In the context of conversational dialogs, transfer learning involves training a model on a large dataset of conversational data to learn general patterns of language use and then fine-tuning the model on a smaller, task-specific dataset to improve the performance for that task; a similar process is detailed in [64]. The following are the steps involved in the detailed process of extractive QA used in this study:

- **Dataset preprocessing:** We utilized large datasets of questions and their corresponding answers, together with a large corpus collection of textual documents, that contain the contexts in which these questions and answers were taken. During this step, we implemented various pre-reprocessing steps to remove any unwanted elements such

as special characters, stop words, or noisy words. Furthermore, we cleaned the corpus by removing any irrelevant or misleading information.

- **Initialization:** In this study, we used several pre-trained transformers. The final fully connected layer(s) of the pre-trained network were removed and replaced with new layer(s) that represent the questions/queries and responses/answers. This process saved a lot of time and computational resources compared to training a network from scratch, as the network can start from a good initial state based on its prior experience. Several parameters were initialized, such as the patch size, the number of epochs, and the learning rate, wherein we used initialization settings similar to those of state-of-the-art studies in QA tasks in English.

- **Fine-tuning:** Several BERT-like transformers in Arabic were fine-tuned using large datasets of annotated QA pairs for the task of extractive QA. This step was crucial to achieving the aims of our study, whereby the goal of the model was to read a passage of text and extract a concise answer to a given question from the passage. To elaborate, we first provided the model with a dataset of questions and their corresponding answers, as well as the passage from which the answer was extracted. The model was then trained to predict the correct answer given a passage and a question. During the fine-tuning process, the transformers' last (i.e., added) layers were trained using a task-specific loss function that aimed to optimize the model to generate the correct answer for a given question. The model was trained to select the answer by identifying the start and end positions of the answer in the passage. The fine-tuning process involved adjusting the weights of the pre-trained transformers using backpropagation to optimize the model's output. The loss function was minimized in several epochs to improve the model's accuracy in predicting the correct answer to a given question.

- **Answer extraction and evaluation:** Once the fine-tuning was complete, our models were used for extractive QA in an Arabic chatbot. When a user asks a question, the chatbot can feed the question into the model, which will then provide an answer based on corpus collection with a confidence score. Hence, our proposed models were tested on different datasets and real-world scenarios to check their robustness and accuracy. In order to compute the confidence and similarity scores, several semantic embedding models were used. A semantic embedding model is an NLP method that allows words or phrases to be represented as vectors of numbers in a multi-dimensional space. The idea behind this model is that words that are similar in meaning will be located close to each other in this space, while words that are dissimilar will be located far apart. In this study, variants of distilbert and BERT-based models for Arabic were employed to predict the answers or responses based on their surrounding context.

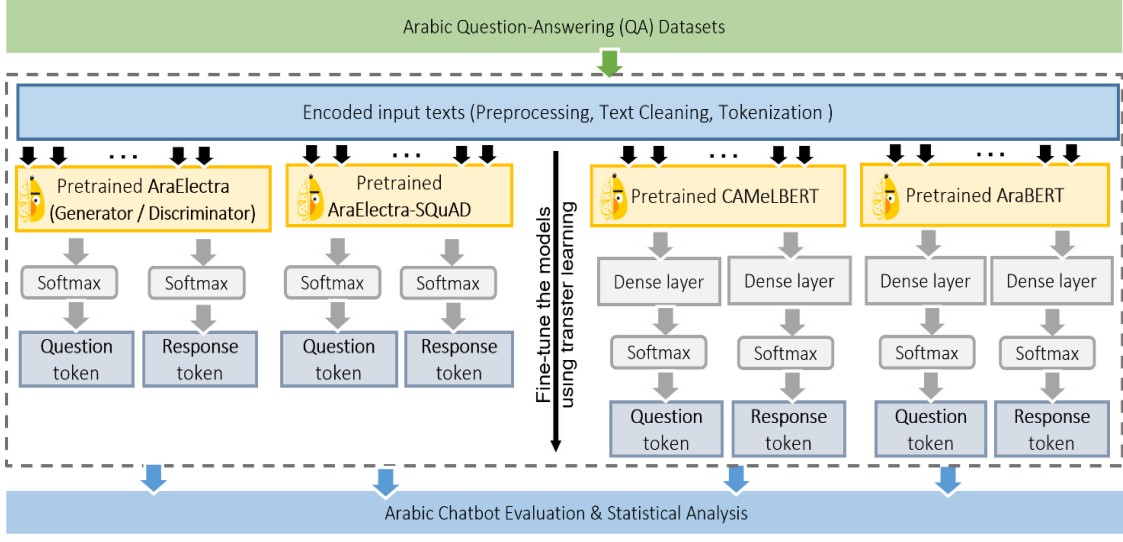

**Figure 2.** General framework of the methodology used in this study.

A Python-style pseudo-code for extractive QA and a transfer learning method for the Arabic chatbot are shown in Algorithm 1.

---

**Algorithm 1** Transfer learning and extractive QA method for Arabic chatbot

---

*# Input:*
*INDEXDIR   : path to the index directory that contains the documents from which answers are extracted*
*QUESTIONS: dataset of questions to be evaluated*

*# Models:*
*_model(str)        : name of BERT/SQUAD model to be fine-tuned on the corpus dataset.*
*_emb_model(str): name of BERT model to use to generate embeddings for semantic similarity.*

*# Output:*
*answer, context, similarity, confidence, doc_index*

*#Stage 1: Preprocessing (text cleaning, tokenization)*
*foreach doc in dataset:*
    *doc = remove_punctuations(doc)*
    *doc = remove_diacritics(doc)*
    *doc = tokenizer.from_pretrained(_emb_model, doc)*
    *save(doc, dataset) in INDEXDIR*

*#Stage 2: Hyperparameters initialization*
*batch_size(int)          : number of question-context pairs fed to model at each iteration*
*n_docs_retrieved(int): number of top relevant documents that will be searched for answer*
*n_answers(int)           : maximum number of candidate answers to return*
*epochs(int)              : number of iterations*
*learning_rate(float): learning rate during training (fit) the model*

*#Stage 3: Fine-tuning: implementing transfer learning*
*base_model = _model*
*if base_model is AraBERT or CAMeLBERT:*
    *model = sequential()*
    *model.add(base_model)*
    *model.flatten()*
    *model.add(dense_layer)*
    *model.add(softmax_layer)*
    *model.compile()*
    *model.fit(dataset in INDEXDIR) # minimize loss function in several epochs to improve the accuracy*
*elseif base_model is AraElectra-SQuAD or AraElectra Generator/Discriminator:*
    *model = sequential()*
    *model.add(base_model)*
    *model.flatten()*
    *model.add(softmax_layer)*
    *model.compile()*
    *model.fit(dataset in INDEXDIR) # minimize loss function in several epochs to improve the accuracy*

*#Stage 4: Answer extraction: submit question to obtain candidate answers.*
*question(str): question in the form of a string*
*foreach question in QUESTIONS:*
    *answer, context, doc_index = get_answer(question, model)*
    *confidence = P(answer | question,context) * P(question | context)/P(answer)*
    *similarity =(question·answer.T)/(norm(question)* norm(answer))*

---

BERT (bidirectional encoder representations from transformers) is known for its ability to capture a deeper understanding of the context of language by training a model to predict missing words in each sentence. This allows BERT to learn contextual relationships between words and provide more accurate and relevant results for NLP tasks. In this study, several

BERT-like transformers were developed for the Arabic language, and they can be divided into two sets: the first set contains several variants from AraBERT and CAMeLBERT transformers that were pre-trained for general text predictions tasks, and the second set contains variants from AraElectra-SQuAD and AraElectra (generator/discriminator) transformers that were pre-trained for text discrimination and QA tasks. The AraBERT is a BERT model for Arabic that has achieved state-of-the-art performance on a range of NLP tasks and has become an important tool for many researchers and practitioners in the field. In this study, we used the base models and the large models, which also differ in terms of the pre-segmentation techniques used. The CAMeLBERT models are pre-trained BERT models for Arabic, including modern standard Arabic (MSA), dialectal Arabic (DA), and classical Arabic (CA), in addition to a model pre-trained on a mix of the three. These models can provide high-quality contextualized word representations for Arabic text.

On the other hand, models that are variants of the Electra model, which stands for Efficiently Learning an Encoder that Classifies Token Replacements Accurately, were pre-trained to perform well on the Stanford Question Answering Dataset (SQuAD). The AraElectra-SQuAD transformer is a language model for Arabic language comprehension that is pre-trained to read passages and generate the correct answers to given questions. The AraElectra-SQuAD transformer was fine-tuned specifically on Arabic SQuAD data to excel at QA tasks in Arabic. Other variants of Electra based models are the AraElectra-base generator and the AraElectra-base discriminator, developed by the team at the AUB Mind Lab; these models differ in terms of their underlying architecture and purpose. The generator is a generative model trained on large amounts of Arabic text data that can generate coherent and contextually relevant text based on a given prompt. It is designed to generate new text that is similar in style and content to the data it was trained on. The discriminative model is a classifier trained to classify Arabic texts into different categories based on the examples it was trained on. It is designed to make predictions or decisions based on input features. Table 3 summarizes the variants of the transformers used in this study in terms of their size, task, description, and pre-training datasets.

**Table 3.** Summary of recent transformers pre-trained for text generation and question answering tasks for Arabic language.

| Transformer Name (*Based on Huggingface*) | Size | Task | Description | Pre-Training Datasets |
|---|---|---|---|---|
| aubmindlab/bert-base-arabertv02 aubmindlab/bert-base-arabertv2 aubmindlab/bert-base-arabertv01 aubmindlab/bert-base-arabert (https://huggingface.co/aubmindlab/bert-base-arabert (**accessed on** 3 March 2023)) | base | Text generation | AraBERT is a pre-trained Arabic language model with pre-segmented text, trained and evaluated similarly to the original BERT in English. | OSCAR, Arabic Wikipedia, Arabic Books collected from various sources, Arabic News Articles and Arabic text collected from social media platforms, such as Twitter and online forums. |
| aubmindlab/bert-large-arabertv2 aubmindlab/bert-large-arabertv02 | large | Text generation | | |
| aubmindlab/araelectra-base-generator (https://huggingface.co/aubmindlab/araelectra-base-generator (**accessed on** 3 March 2023)) | base | Text prediction, QA | This generator model, which generates new text based on learned patterns from training data, achieved state-of-the-art performance on Arabic QA datasets. | OSCAR unshuffled and filtered, Arabic Wikipedia dump from 1 September 2020, the 1.5 B words Arabic Corpus, the OSIAN Corpus, and Assafir news articles. |
| aubmindlab/araelectra-base-discriminator (https://huggingface.co/aubmindlab/araelectra-base-discriminator (**accessed on** 3 March 2023)) | base | Text prediction, QA | This discriminator model classifies or makes predictions based on input features. | |

**Table 3.** *Cont.*

| Transformer Name (*Based on Huggingface*) | Size | Task | Description | Pre-Training Datasets |
|---|---|---|---|---|
| CAMeL-Lab/bert-base-arabic-camelbert-mix (https://huggingface.co/CAMeL-Lab/bert-base-arabic-camelbert-mix (accessed on 3 March 2023)) CAMeL-Lab/bert-base-arabic-camelbert-ca CAMeL-Lab/bert-base-arabic-camelbert-da CAMeL-Lab/bert-base-arabic-camelbert-msa | base | Text generation | Pre-trained BERT models for Arabic texts with different dialects and structures, formal and informal Arabic. | MSA: Arabic Gigaword, Abu El-Khair Corpus, OSIAN corpus, Arabic Wikipedia, Arabic OSCAR DA: A collection of dialectal data CA: OpenITI (Version 2020.1.2). |
| ZeyadAhmed/AraElectra-Arabic-SQuADv2-QA (https://huggingface.co/ZeyadAhmed/AraElectra-Arabic-SQuADv2-QA (accessed on 3 March 2023)) | base | QA | AraElectra-based model fine-tuned on QA pairs to predict unanswerable questions. | Arabic-SQuADv2.0 dataset. |

## 4. Experimental Setup

### 4.1. Datasets

In order to evaluate the proposed methods, it is necessary to fine-tune the transformers on a corpus collection of Arabic texts and then to evaluate their confidence in answering questions using a set of Arabic questions. Thus, the datasets utilized in this study contain both a corpus of texts and a set of questions. One dataset was created by Aljawarneh [65]; it contains 398 Arabic questions generated using augmentation techniques. The questions are presented in MSA, but it has no corpus collection of texts. Therefore, we collected a corpus of 398 documents from the web about the topics found in the question set. We referred to this dataset as MSA-QA. We also used the Arabic language comprehension dataset (ARCD) created by Mozannar et al. [66]. This dataset, which we refer to as ARCD-QA, comprises 1395 distinct questions created by crowd workers from articles on Arabic Wikipedia. Each question is accompanied by the corresponding article title, the context in which the question was raised, and a set of potential answers. The corpus collection we used with this dataset is the Arabic Wikipedia dump 2021, which originally contained 600, 000 documents from Wikipedia. However, due to limitations related to computing power, 365,568 documents were indexed and used in our study. The third dataset is the Arabic AskFM dataset, which comprised 98,422 question–answer pairs from the AskFM platform posted in dialectal (informal) Arabic (mostly Egyptian dialects), and we refer to this dataset as DA-QA. The questions in this dataset focus on Islamic topics, and we used this dataset in some of our initial experiments. To gain insights into these datasets, we utilized several metrics such as word and character counts, recognition of frequent queries and terms, visual depictions of word frequencies, and word occurrence analysis. Table 4 summarizes the datasets used in this study.

### 4.2. Resources and Tools

The resources utilized in this research include two PCs, each with an Intel(R) Core(TM) i7-10700T CPU with 2.00 GHz and 16.0 GB RAM. Model fine-tuning and testing were implemented using TensorFlow 2.8.0, which uses Keras as a high-level API with various complementary libraries such as Ktrain, Scikit-Learn, Matplotlib, and Pandas. The first batch of experiments, using 10 questions, took around 10–20 min each. The second batch of experimental works, using ~400 questions, took around 2–4 h each based on the transformers, while the third batch, using ~1440 questions, took around 20 h each.

### 4.3. Evaluation

In natural language processing (NLP), the confidence metric is a score that measures the level of certainty or the probability of a model to accurately predict or classify the correct label or outcome of a given text sample. It is essentially a way to measure the quality of the predicted result. The confidence metric is often expressed as a value between 0 and 1. A higher confidence score indicates that the model is more certain about its

prediction, while a lower score indicates greater uncertainty. The confidence metric is used for answer predictions, where the accuracy of the model's prediction is important. There are several mathematical equations used to compute confidence in QA research, depending on the specific approach and model used. In this research, we used the confidence score shown below:

$$Conf = P(answer|question,\ context) * P(question\ |\ context)/P(answer)$$

where $P(answer\ |\ question,\ context)$ is the probability of the answer given the question and context, $P(question\ |\ context)$ is the probability of the question given the context, and $P(answer)$ is the prior probability of the answer. This equation calculates the probability that a given answer is correct, considering both the likelihood of the answer given the question and context, and the frequency of the question and answer in the corpus collection. Another metric that is often used is similarity, which evaluates the semantic similarity between two texts of the question and the answer. These metrics are used to assess the relevance and correctness of the answers generated by the chatbot and how well an answer captures the relevant information from the given question, which was calculated in this study as follows:

$$Sim = (question \cdot answer \cdot T)/(norm(question) * norm(answer))$$

wherein the vectors representing the *question* and the *answer* to be compared were used, *T* is the transpose of the *answer* vector, and the *norm* is value of matrix norm computed in *NumPy Python*.

**Table 4.** Summary of the datasets for Arabic question answering (QA) and Arabic chatbots used in this study.

| Dataset | Number of Documents | | | Description |
| | Questions | Answers | Corpus | |
| --- | --- | --- | --- | --- |
| **MSA-QA** | 398 | 398 | 398 | This repository of Arabic Questions Dataset (https://github.com/EmranAljawarneh/Arabic-questions-dataset (accessed on 25 April 2023)) provides an Arabic question for data science and machine learning. |
| **ARCD-QA** | 1395 | 1395 | 365,568 | The corpus contains a comprehensive Arabic Wikipedia dump 2021 (https://www.kaggle.com/datasets/z3rocool/arabic-wikipedia-dump-2021?datasetId=1179369 (accessed on 1 May 2023)), including articles, discussions, and textual information from 2021. The questions were created by crowd-workers in ARCD (https://www.kaggle.com/datasets/thedevastator/unlocking-arabic-language-comprehension-with-the (accessed on 1 May 2023)). |
| **DA-QA** | 98,422 | 98,422 | 98,422 | Arabic AskFM dataset collection of questions and answers mostly about Islamic topics by various authors in dialectal Arabic (DA) on the AskFM platform. |

## 5. Experimental Results

*5.1. Initial Results Using a Sample of Selected Questions from the MSA-QA and DA-QA Datasets*

For the initial investigation, the selected questions were tested for modern standard Arabic and dialectal (i.e., informal) Arabic using the proposed transformers. Tables 5 and 6 show the experimental results obtained from 10 questions chosen from the MSA-QA and DA-QA datasets, respectively. In both tables, we fine-tuned one of the AraBERT transformers (*bert-base-arabert*) and the AraElectra-SQuAD transformer using two sizes of semantic embedding models (*bert-base-arabert* and *bert-large-arabert*v2). For each given question shown in the tables, we extracted the answer and then evaluated the similarity and confidence of each model in giving such an answer. The confidence scores that exceeded 0.8

are presented in bold. While the semantic similarity scores are mostly high, the confidence values indicate that the fine-tuned models perform well with some questions and not with others. To offer a broad view of these results, Figure 3 visualizes the confidence scores, and Figure 4 visualizes the similarity scores obtained by the AraBERT and AraElectra-SQuAD transformers using the selected questions from the MSA-QA and DA-QA datasets. As shown in the figures, the similarities do not indicate differences between the models because there was relevant information for each given question in the texts on which the models were fine-tuned. We found a slight drop in the similarity scores when we used AraElectra-SQuAD with the large semantic embedding model. On the other hand, we found that confidence metrics effectively indicated the differences in the models' performance. Figure 5 offers a more detailed view of the results of the confidence scores obtained by the AraElectra-SQuAD transformer using the selected questions from MSA-QA, indicating the formal context of Arabic, and the questions from DA-QA, indicating the informal context of Arabic. We found that the AraElectra-SQuAD transformer performs well for the formal contexts but obtained lower scores for the informal questions. Another notable finding is that the confidence of the model does not change when the semantic embedding models are changed, and the only change occurred in the similarity results.

**Table 5.** Experimental results of 10 selected questions from modern standard Arabic (MSA-QA) dataset.

| Question | AraBERT | | | | AraElectra-SQuAD | | | |
| | Base | | Large | | Base | | Large | |
| | Sim. | Conf. | Sim. | Conf. | Sim. | Conf. | Sim. | Conf. |
|---|---|---|---|---|---|---|---|---|
| 1-(؟ علم البيانات ؟') | 0.8890 | 0.3355 | 0.9284 | 0.1700 | 0.9191 | 0.9993 | 0.6480 | 0.9993 |
| 2-('تعريف البيانات العلمية؟') | 0.9447 | 0.1249 | 0.9486 | 0.1867 | 0.9652 | 0.8601 | 0.6321 | 0.8601 |
| 3-('اذكر التباين بين علم البيانات والذكاء الاصطناعي؟ ؟') | 0.9980 | 1.0 | 0.9291 | 0.1234 | 0.9801 | 0.5006 | 0.7498 | 0.5006 |
| 4-('أذكر لغات جداول الأعمال الأكثر شيوعًا في مجال علم البيانات؟ ؟') | 0.9905 | 0.5 | 0.9531 | 0.1741 | 0.9778 | 0.9276 | 0.8334 | 0.9276 |
| 5-('يعرف علم البيانات؟') | 0.9424 | 0.25 | 0.9384 | 0.2627 | 0.9343 | 0.9951 | 0.6901 | 0.9951 |
| 6-('اذكر مجالات العمل التي يشارك فيها علم البيانات؟') | 0.9585 | 0.1250 | 0.9895 | 0.5 | 0.9747 | 0.5020 | 0.8945 | 0.5020 |
| 7-('أذكرلي لغات البرمجة الأكثر شيوعًا في مجال علم البيانات؟ ؟') | 0.9908 | 0.1702 | 0.9887 | 0.1184 | 0.9847 | 0.2424 | 0.8175 | 0.2424 |
| 8-('أذكر الاختلافات بين علم البيانات والذكاء الاصطناعي؟ ؟') | 0.9670 | 1.0 | 0.9419 | 0.5476 | 0.9817 | 0.4718 | 0.8070 | 0.4718 |
| 9-('أذكر الفرق بين علم البيانات والمثقفين الاصطناعي؟ ؟') | 0.9851 | 0.5102 | 0.9453 | 0.2202 | 0.9831 | 0.9954 | 0.8256 | 0.9954 |
| 10-('اذكر مجالات العمل التي تدخل فيها البيانات علميا؟ ؟') | 0.9470 | 0.5 | 0.9677 | 0.1448 | 0.9686 | 0.3397 | 0.8673 | 0.3397 |

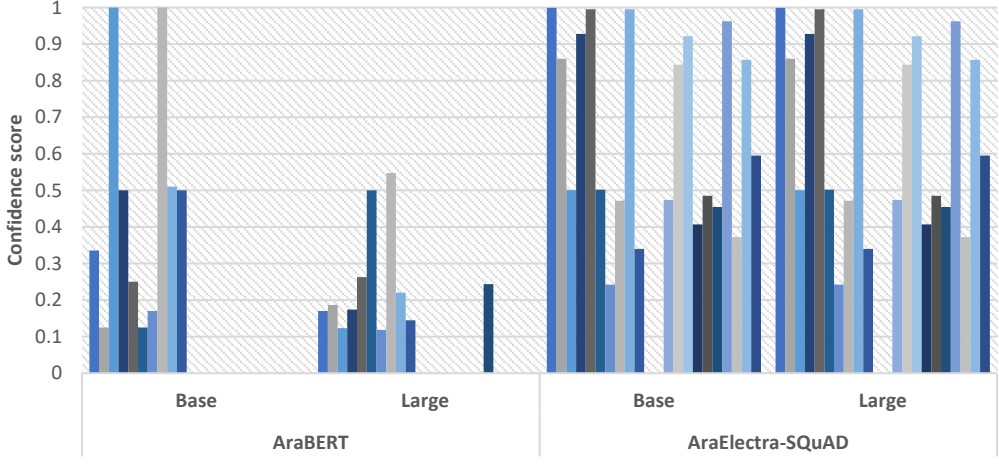

**Figure 3.** Confidence scores obtained by AraBERT and AraElectra-SQuAD transformers using base and large sizes of embedding models on a sample of selected questions from MSA-QA and DA-QA datasets.

**Table 6.** Experimental results of 10 selected questions from dialectal Arabic (DA-QA) dataset.

| Question | AraBERT | | | | AraElectra-SQuAD | | | |
| --- | --- | --- | --- | --- | --- | --- | --- | --- |
| | Base | | Large | | Base | | Large | |
| | Sim. | Conf. | Sim. | Conf. | Sim. | Conf. | Sim. | Conf. |
| 1-(؟هوا انا ينفع اقول اني بحب ربنا اووي عشان هوا عسل وبيحبنا؟') | 0.9825 | 0.2331 | 0.9366 | 0.1322 | 0.9729 | 0.4737 | 0.8815 | 0.4737 |
| 2-(؟بالله عليك اجنبي هل يجوز للحائض زيارة المقابر ضروري بالله عليك؟') | 0.9801 | 0.1713 | 0.9848 | 0.1724 | 0.9710 | 0.8436 | 0.7874 | 0.8436 |
| 3-(؟'التنورة وفوقها بلوزة طويلة مع طرحة تغطي الصدركده حجاب شرعي؟') | 0.9902 | 0.4014 | 0.9603 | 0.2438 | 0.9767 | 0.9217 | 0.8508 | 0.9217 |
| 4-(؟ هل يجوز أن أقبل يد اخى الأكبركشكر وعرفان لفضله عليا منذ صغرى ؟') | 0.9809 | 0.2571 | 0.9936 | 0.3674 | 0.9827 | 0.4069 | 0.9191 | 0.4069 |
| 5-(؟؟هل في سنه عن النبي اننا لما نمسح الارض بالمياه نحط عليها ملح ؟؟') | 0.9858 | 0.1859 | 0.975 | 0.1683 | 0.9762 | 0.4853 | 0.8186 | 0.4853 |
| 6 - (؟'هل ترى تقسيم البدعة الى حسنة وسيئة؟ ') | 0.9842 | 0.3012 | 0.9371 | 0.2435 | 0.9675 | 0.4547 | 0.6798 | 0.4547 |
| 7-(') ماما بتسأل حضرتك يا شيخ لو هي شارية ليا حاجات للمستقبل أدوات منزلية وغيره هل عليها زكاة أم لا ؟ ') | 0.986 | 0.3289 | 0.9763 | 0.1594 | 0.9838 | 0.9624 | 0.8619 | 0.9624 |
| 8-(') طب يا شيخ بالنسبة لاعياد الميلاد؟ ') | 0.9846 | 0.3816 | 0.946 | 0.1539 | 0.9584 | 0.3724 | 0.7328 | 0.3724 |
| 9-(') يا شيخنا هل صلاة المسجد للرجال فرض وتاركه آثم غير مقبول صلاه ؟ ') | 0.9408 | 0.2420 | 0.9839 | 0.1719 | 0.9764 | 0.8567 | 0.8407 | 0.8567 |
| 10-(') هل يمكن لتوبه صادقه ان تمحو ما قبلها فكأنما ما أذنب المرء قط ؟!') | 0.9469 | 0.4698 | 0.9851 | 0.1635 | 0.9792 | 0.5950 | 0.8337 | 0.5950 |

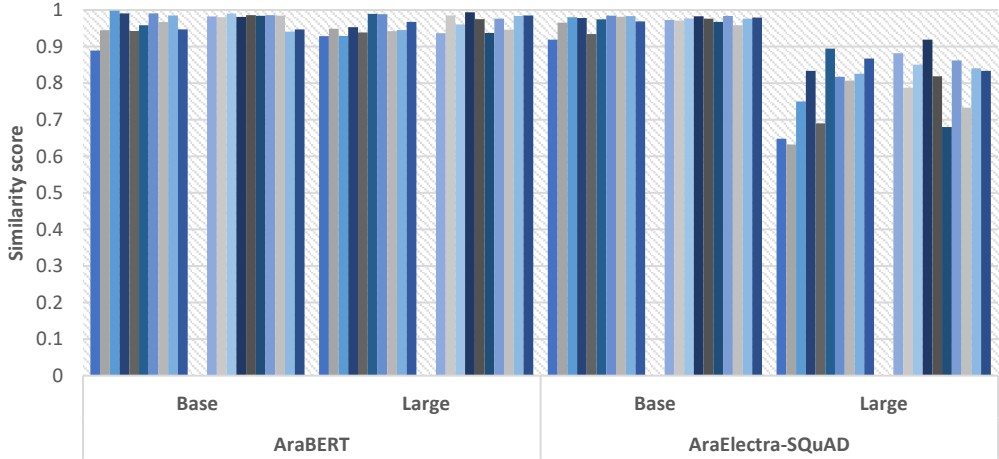

**Figure 4.** Similarity scores obtained by AraBERT and AraElectra-SQuAD transformers using base and large sizes of embedding models on a sample of selected questions from MSA-QA and DA-QA datasets.

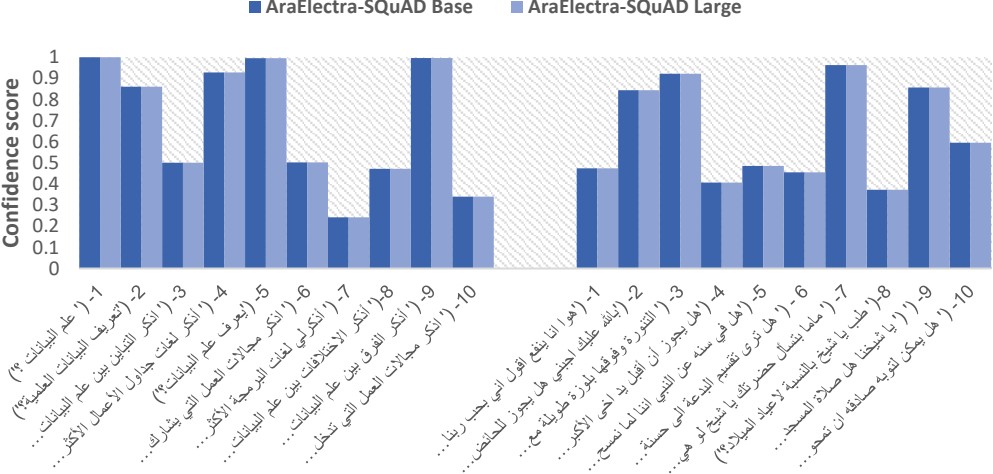

**Figure 5.** Comparison between confidence scores obtained by formal and informal questions using AraElectra-SQuAD transformer.

### 5.2. Initial Results Using a Sample of Selected Questions from the MSA-QA and ARCD-QA Datasets

To extend our experiments, we aimed in this set of experiments to compare MSA-QA and ARCD-QA using a sample of questions that were generated randomly; we investigated the answers using the proposed transformers, as well as their confidence and similarity scores. In Table 7, we used ten variants of AraBERT-based models with each dataset sample. For MSA-QA, the top-performing model was base-arabic-camelbert-da, with an average confidence of 0.4612 and an average similarity of 0.8168. Closely behind was the bert-large-arabertv02 transformer, resulting in an average confidence of 0.4504 and an average similarity of 0.6657. For ARCD-QA, the top-performing models were bert-base-arabert and bert-large-arabertv02, with 0.48 confidence scores. As shown in Table 8, multiple AraElectra-based models were utilized in the experiment, including the AraElectra-SQuAD and AarElectra generator and discriminator. We found that the AraElectra-Arabic-SQuADv2-QA with the distilbert-base-uncased and bert-base-arabertv2 embedding models outperformed other models on both datasets. The confidences scores obtained in this set of experiments are visualized in Figure 6 for the MSA-QA sample and Figure 7 for the ARCD-QA sample.

**Table 7.** Experimental results of a sample of generated questions from MSA-QA and ARCD-QA datasets using AraBERT-based transformers.

| Dataset | Transformer | Semantic Embeddings Model | Avg. Sim. | Avg. Conf. |
|---|---|---|---|---|
| **MSA-QA** | bert-base-arabertv02 | bert-base-arabertv02 | 0.8457 | 0.3304 |
| | bert-large-arabertv02 | bert-large-arabertv02 | 0.6657 | 0.4504 |
| | bert-base-arabertv2 | bert-base-arabertv2 | 0.8915 | 0.1695 |
| | bert-large-arabertv2 | bert-large-arabertv2 | 0.7727 | 0.3989 |
| | bert-base-arabertv01 | bert-base-arabertv01 | 0.8183 | 0.2779 |
| | bert-base-arabert | bert-base-arabert | 0.7776 | 0.2452 |
| | bert-base-arabic-camelbert-mix | bert-base-arabic-camelbert-mix | 0.47667 | 0.3876 |
| | bert-base-arabic-camelbert-ca | bert-base-arabic-camelbert-ca | 0.9625 | 0.1935 |
| | bert-base-arabic-camelbert-da | bert-base-arabic-camelbert-da | 0.8168 | 0.4612 |
| | bert-base-arabic-camelbert-msa | bert-base-arabic-camelbert-msa | 0.5394 | 0.2493 |
| **ARCD-QA** | bert-base-arabertv02 | bert-base-arabertv02 | 0.7599 | 0.3320 |
| | bert-large-arabertv02 | bert-large-arabertv02 | 0.6774 | 0.4816 |
| | bert-base-arabertv2 | bert-base-arabertv2 | 0.6491 | 0.1822 |
| | bert-large-arabertv2 | bert-large-arabertv2 | 0.6519 | 0.2913 |
| | bert-base-arabertv01 | bert-base-arabertv01 | 0.8635 | 0.2271 |
| | bert-base-arabert | bert-base-arabert | 0.8507 | 0.4800 |
| | bert-base-arabic-camelbert-mix | bert-base-arabic-camelbert-mix | 0.9122 | 0.2598 |
| | bert-base-arabic-camelbert-ca | bert-base-arabic-camelbert-ca | 0.9352 | 0.2972 |
| | bert-base-arabic-camelbert-da | bert-base-arabic-camelbert-da | 0.8664 | 0.2937 |
| | bert-base-arabic-camelbert-msa | bert-base-arabic-camelbert-msa | 0.7378 | 0.3381 |

**Table 8.** Experimental results of a sample of generated questions from MSA-QA and ARCD-QA datasets using AraElectra-based transformers.

| Dataset | Transformer | Semantic Embeddings Model | Avg. Sim. | Avg. Conf. |
|---------|-------------|---------------------------|-----------|------------|
| **MSA-QA** | AraElectra-Arabic-SQuADv2-QA | bert-base-arabertv2 | 0.8242 | 0.6675 |
| | AraElectra-Arabic-SQuADv2-QA | distilbert-base-uncased | 0.9786 | 0.6675 |
| | araelectra-base-generator | bert-base-arabertv2 | 0.6434 | 0.4179 |
| | araelectra-base-discriminator | bert-base-arabertv2 | 0.7652 | 0.4329 |
| | araelectra-base-generator | distilbert-base-uncased | 0.9687 | 0.3043 |
| | araelectra-base-discriminator | distilbert-base-uncased | 0.5688 | 0.4286 |
| **ARCD-QA** | AraElectra-Arabic-SQuADv2-QA | bert-base-arabertv2 | 0.6952 | 0.6116 |
| | AraElectra-Arabic-SQuADv2-QA | distilbert-base-uncased | 0.9806 | 0.6116 |
| | araelectra-base-generator | bert-base-arabertv2 | 0.7385 | 0.1957 |
| | araelectra-base-discriminator | bert-base-arabertv2 | 0.7388 | 0.2086 |
| | araelectra-base-generator | distilbert-base-uncased | 0.9166 | 0.3206 |
| | araelectra-base-discriminator | distilbert-base-uncased | 0.8962 | 0.4593 |

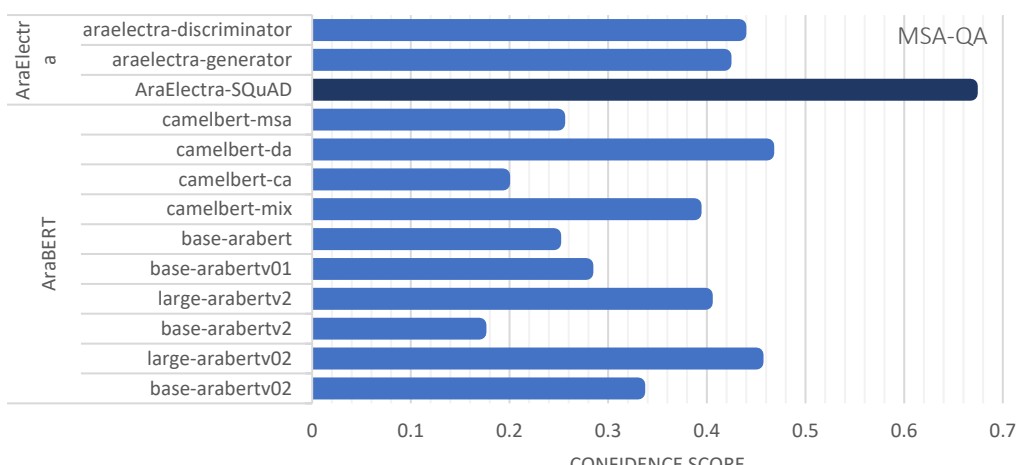

**Figure 6.** Comparison between confidence scores obtained by AraBERT- and AraElectra-based transformers on a sample of selected questions from MSA-QA dataset.

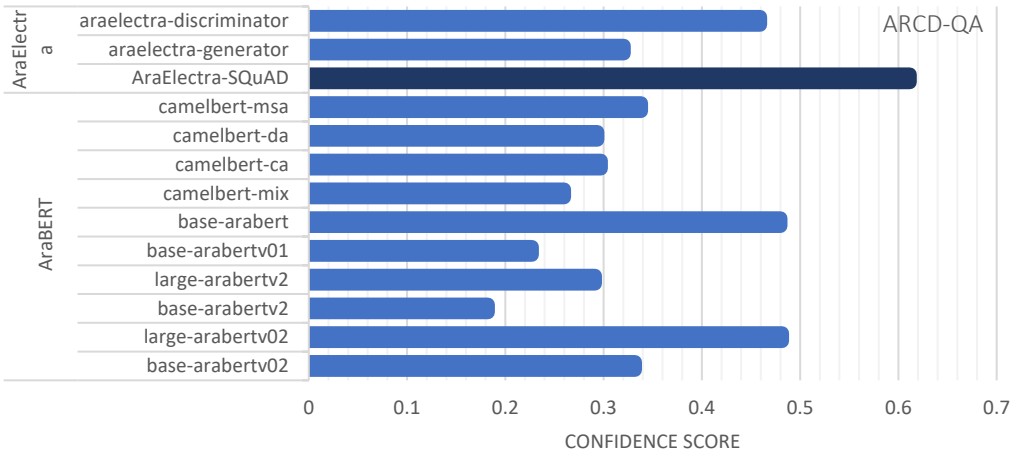

**Figure 7.** Comparison between confidence scores obtained by AraBERT- and AraElectra-based transformers on a sample of selected questions from ARCD-QA dataset.

## 5.3. Experimental Results Using All Questions from the MSA-QA and ARCD-QA Datasets

In the final set of our experiments, we used all of the dataset questions and indexed collections specified previously in Table 4. A number of models were selected based on the experiments conducted in the previous parts, as described in Sections 5.1 and 5.2. These models were selected for evaluation based on their confidence results and the robustness of their performance. Table 9 and Figure 8 show our Arabic chatbot evaluation results using 398 questions in MSA-QA with their collected corpus, and 1395 questions in ARCD-QA with 365,568 indexed Wikipedia documents. Significant findings include the result that the AraElectra-based fine-tuned models outperformed the AraBERT-based fine-tuned models, and the AraElectra-SQuAD model was the model that performed best for all datasets. Its best confidence and similarity scores were achieved when using *distilbert* semantic embeddings, with results of 0.9660 similarity and 0.6658 confidence. Therefore, the AraElectra-SQuAD model can be further enhanced in fine-tuning and in practice in various natural language processing tasks, such as chatbots, virtual assistants, and information retrieval systems, for Arabic-speaking users. By combining the power of the transformer architecture with the fine-tuning on SQuAD, AraElectra-SQuAD was able to provide accurate and contextually relevant answers to questions in Arabic.

**Table 9.** Arabic chatbot evaluation results using all questions in MSA-QA, and ARCD-QA with Wikipedia dump collection.

| | Dataset | Transformer | Semantic Embeddings Model | Avg. Sim. | Avg. Conf. |
|---|---|---|---|---|---|
| **AraBERT-based** | MSA-QA | bert-base-arabertv02 | bert-base-arabertv02 | 0.8256 | 0.3897 |
| | | bert-large-arabertv02 | bert-large-arabertv02 | 0.8365 | 0.2128 |
| | | bert-large-arabertv2 | bert-large-arabertv2 | 0.7673 | 0.4251 |
| | | bert-base-arabic-camelbert-da | bert-base-arabic-camelbert-da | 0.9229 | 0.3634 |
| | ARCD-QA | bert-base-arabertv02 | bert-base-arabertv02 | 0.6986 | 0.2038 |
| | | bert-large-arabertv02 | bert-large-arabertv02 | 0.6241 | 0.5465 |
| | | bert-base-arabert | bert-base-arabert | 0.9396 | 0.2426 |
| | | bert-base-arabic-camelbert-msa | bert-base-arabic-camelbert-msa | 0.7727 | 0.1901 |
| **AraElectra-based** | MSA-QA | AraElectra-Arabic-SQuADv2-QA | bert-base-arabertv2 | 0.8268 | 0.6422 |
| | | AraElectra-Arabic-SQuADv2-QA | distilbert-base-uncased | 0.9773 | 0.6422 |
| | | araelectra-base-generator | bert-base-arabertv2 | 0.7013 | 0.3616 |
| | | araelectra-base-discriminator | bert-base-arabertv2 | 0.7218 | 0.3291 |
| | ARCD-QA | AraElectra-Arabic-SQuADv2-QA | bert-base-arabertv2 | 0.6852 | 0.6657 |
| | | AraElectra-Arabic-SQuADv2-QA | distilbert-base-uncased | 0.9660 | 0.6658 |
| | | araelectra-base-generator | distilbert-base-uncased | 0.9036 | 0.2908 |
| | | araelectra-base-discriminator | distilbert-base-uncased | 0.8573 | 0.4147 |

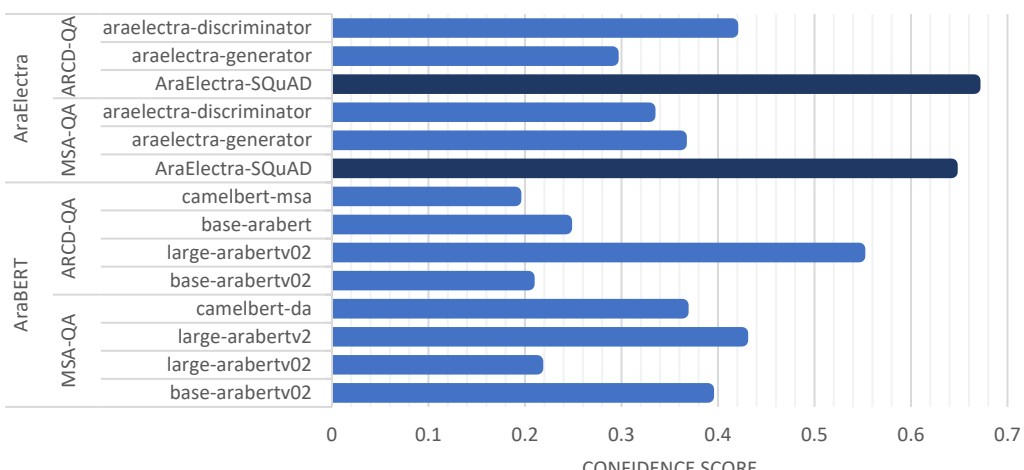

**Figure 8.** Comparison between confidence scores obtained by AraBERT and AraElectra-SQuAD transformers on a sample of selected questions from ARCD-QA dataset.

## 6. Conclusions and Future Works

Chatbots are AI-based programs designed to imitate human dialog. This paper addresses the problem of Arabic chatbots because they are rare and less well known than English chatbots due to the nature and intricacy of the Arabic language. This work offers a comprehensive review of previously published studies that applied chatbots using extractive and generative models, machine learning, deep learning, and the current trends in transfer learning. Using pre-trained models and transfer learning, we overcame the problems of limited data availability and allowed for the generalization of Arabic-language conversational dialogs, enabling the model to understand the conversational context and behave naturally. Different Arabic QA datasets were utilized to investigate the use of transfer learning techniques using ten AraBERT-based and CAMeLBERT transformers, as well as six AraElectra-SQuAD and AarElectra generator and discriminator transformers. We evaluated different variants of these transformers and semantic embedding models using a dataset with 398 questions and corresponding documents and another with 1395 questions and 365,568 documents indexed from Arabic Wikipedia. Through extensive experimentation, we observed that the AraElectra-based fine-tuned models yielded promising results with both datasets. The AraElectra-Arabic-SQuADv2-QA model consistently demonstrated the best performance, with 0.66 confidence and 0.96 similarity scores. Several limitations exist in this work, including the need for more efficient computational resources for indexing documents, the difficulty of pre-processing Arabic texts, and the large size of the downloaded Arabic transformers. Another significant limitation is that we could not mix both extractive and generative QA in this evaluation study because generative and extractive methods have different structures, purposes, and domains. In addition, they have different evaluation measures. The evaluation metrics for extractive QA used in this study are NLP metrics: confidence and similarity. These metrics cannot be used with generative QA because generative AI approaches are evaluated by humans rating the quality of the generated output based on various criteria, such as creativity, coherence, and overall quality; perplexity to measure how well the model predicts the next word in each sequence of words; and diversity to measure how diverse the generated output is. Other metrics that are specific to generative types of models include BLEU or ROUGE scores for language generation models. P

For future work, it is strongly recommended that this work be continued to compare different generative AI models for Arabic chatbots and evaluate them using humans' ratings and automated metrics such as coherence, quality, perplexity, diversity, BLEU, and ROUGE. It would also be beneficial to explore additional datasets to validate the findings, assess the models' performance in diverse contexts, and improve the results in terms of confidence. Additionally, evaluating the models' performance in other language-related

tasks or expanding the research to include multilingual question answering would be valuable directions for future studies.

**Author Contributions:** Conceptualization, T.N.A. and S.M.A.; methodology, T.N.A.; software, T.N.A.; validation, S.M.A.; formal analysis, S.M.A.; investigation, T.N.A.; resources, T.N.A.; writing—original draft preparation, S.M.A.; writing—review and editing, S.M.A.; visualization, S.M.A.; supervision S.M.A. All authors have read and agreed to the published version of the manuscript.

**Funding:** This research received no external funding.

**Informed Consent Statement:** Not applicable.

**Data Availability Statement:** The indexed dataset is available through this link: bit.ly/3pubIc9, accessed on 15 August 2023.

**Acknowledgments:** The researchers would like to acknowledge the Deanship of Scientific Research, Taif University for funding this work.

**Conflicts of Interest:** The authors declare that they have no known competing financial interest or personal relationships that could have appeared to influence the work reported in this paper.

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
