# Peer review of "Evaluation of an Arabic Chatbot Based on Extractive Question-Answering Transfer Learning and Language Transformers"

_ai, doi:10.3390/ai4030035_

Round 1

Reviewer 1 Report

The respected authors presented their work on Arabic Question Answering using ChatBots, focusing on a transformer-based transfer learning approach. Their comparative analysis and statistical presentation were meticulous and comprehensive.

However, the Abstract needs to be more concise, briefly highlighting the research's premise and key results. Additionally, the methods section should be refined, providing the transfer-learning algorithm in pseudocode with clear explanations.

While the research's focus on transformer-based transfer learning is commendable, there is a notable omission regarding Generative-AI approaches, which are currently highly relevant. To address this, the authors should compare their methods with tools like ChatGPT, Google Bard, Bing, or HuggingFace. Such a comparison is essential to pique the interest of readers who frequently use these tools from their web browsers.

I strongly encourage the respected authors to include a comparison of Generative AI tools in their title, highlighting the importance of their research.

Here is an option: Comparative Evaluation of Arabic Chatbots: Extractive Question-Answering Transfer Learning and Language Transformers with Generative AI.

The respected authors are no obligated in anyway to use the exact title

The respected authors' writing style tends to include long sentences, making the manuscript challenging to follow. By breaking these long sentences into smaller ones, the manuscript will become more coherent and easier to understand.

Reviewer 2 Report

1)      The research contributions should be clearly highlighted in bullet points within Introduction section.

2)      Within Table 1, the advantages and disadvantages of each of the studies needs to be highlighted. Only knowing the description (i.e., what is it) does not help critical analysis (i.e., concerns of the readers).

3)      Table 1 highlights several references like 15, 16, 17, 19 etc. These are missing from Table 2. Why is that? What methods were used in 15, 16, 17, 19 and other missing ones? What were the data sources, accuracy, pros. and cons. of them?

4)      Figure 2 provides a generic framework / conceptual diagram of the methodology. However, authors should also intrude a Pseudocode algorithm to portray the methodology for making it more clear.

5)      What were the limitations of this study? Highlight these limitations within the conclusion section (before discussing the future work). In generation, limitations set forth the directions of future research endeavors.

6)      Reference section is missing most recent research works. Within the reference section, please add reference to research works for 2023.

Round 2

Reviewer 1 Report

* The respected authors responded to most comments in a satisfactory manner

* The respected authors claimed that the ChatGPT is outside the scope of their work. I strongly disagree!

* The respected authors must include a Limitation Section where they justify why ChatGPT is out of scope

* The respected authors must include a future direction and link their work to ChatGPT to make it of great value and connected to the fast-pace GPT tools emerging

Reviewer 2 Report

All my previously suggested improvements have been taken into consideration. I am happy this updated version. 
